# The cost and cost-effectiveness of novel tuberculosis vaccines in low- and middle-income countries: A modeling study

**Allison Portnoy** [1] *, **Rebecca A. Clark** [2,3,4], **Matthew Quaife** [2,3,4], **Chathika K. Weerasuriya** [2,3,4], **Christinah Mukandavire** [2,3,4], **Roel Bakker** [2,3,4,5], **Arminder K. Deol** [2,3,4,6], **Shelly Malhotra** [7,8], **Nebiat Gebreselassie** [9], **Matteo Zignol** [9], **So Yoon Sim** [10], **Raymond C. W. Hutubessy** [10], **Inés Garcia Baena** [9], **Nobuyuki Nishikiori** [9], **Mark Jit** [3,4,11], **Richard G. White** [2,3,4‡], **Nicolas A. Menzies** [1,12‡]

1 Center for Health Decision Science, Harvard T.H. Chan School of Public Health, Boston, Massachusetts, United States of America, 2 TB Modelling Group, London School of Hygiene and Tropical Medicine, London, United Kingdom, 3 Centre for the Mathematical Modelling of Infectious Diseases, London School of Hygiene and Tropical Medicine, London, United Kingdom, 4 Department of Infectious Disease Epidemiology, London School of Hygiene and Tropical Medicine, London, United Kingdom, 5 KNCV Tuberculosis Foundation, The Hague, the Netherlands, 6 Coalition for Epidemic Preparedness Innovations, London, United Kingdom, 7 Market Access, Global Alliance for TB Drug Development, New York, New York, United States of America, 8 Global Access, International AIDS Vaccine Initiative, New York, New York, United States of America, 9 Global TB Programme, World Health Organization, Geneva, Switzerland, 10 Department of Immunization, Vaccines and Biologicals, World Health Organization, Geneva, Switzerland, 11 School of Public Health, University of Hong Kong, Hong Kong SAR, China, 12 Department of Global Health and Population, Harvard T.H. Chan School of Public Health, Boston, Massachusetts, United States of America

‡ These authors are joint senior authors on this work.
* aportnoy@mail.harvard.edu

## Abstract

### Background

Tuberculosis (TB) is preventable and curable but eliminating it has proven challenging. Safe and effective TB vaccines that can rapidly reduce disease burden are essential for achieving TB elimination. We assessed future costs, cost-savings, and cost-effectiveness of introducing novel TB vaccines in low- and middle-income countries (LMICs) for a range of product characteristics and delivery strategies.

### Methods and findings

We developed a system of epidemiological and economic models, calibrated to demographic, epidemiological, and health service data in 105 LMICs. For each country, we assessed the likely future course of TB-related outcomes under several vaccine introduction scenarios, compared to a "no-new-vaccine" counterfactual. Vaccine scenarios considered 2 vaccine product profiles (1 targeted at infants, 1 at adolescents/adults), both assumed to prevent progression to active TB. Key economic inputs were derived from the Global Health Cost Consortium, World Health Organization (WHO) patient cost surveys, and the published literature. We estimated the incremental impact of vaccine introduction for a range of health and economic outcomes. In the base-case, we assumed a vaccine price of $4.60 and used

**Data Availability Statement:** Analytic code is available at https://doi.org/10.5281/zenodo.6421372.

**Funding:** This study was funded by the World Health Organization (2020/985800-0 to RGW), the Bill & Melinda Gates Foundation (INV-001754 to RAC, RGW; OPP1084276 to RGW; OPP1135288 to RGW), the Canadian Centennial Scholarship Fund (to RAC), UK Research & Innovation Medical Research Council (MR/N013638/1 to CKW; CCF17-7779 via SET Bloomsbury to RGW), the Wellcome Trust (218261/Z/19/Z to RGW), the National Institutes of Health (1R01AI147321-01 to RGW), European and Developing Countries Clinical Trials Partnership (RIA208D-2505B to RGW), UK Research & Innovation Economic and Social Research Council (ES/P008011/1 to RGW). Members of the funder participated as authors on the study. All authors had the opportunity to access and verify the data, and all authors were responsible for the decision to submit the manuscript for publication.

**Competing interests:** I have read the journal's policy and the authors of this manuscript have the following competing interests: Members of the funder (NG, MZ, SYS, RH, IGB, NN) participated as authors on the study and critically reviewed the analysis, reviewed and revised the manuscript, and approved the final manuscript as submitted. All other authors have declared that no competing interests exist.

**Abbreviations:** AFR, WHO African region; ART, antiretroviral therapy; COVID-19, Coronavirus Disease 2019; DALY, disability-adjusted life-year; DS, drug-susceptible; GDP, gross domestic product; GLM, generalized linear regression model; GNI, gross national income; HPV, human papillomavirus; ICER, incremental cost-effectiveness ratio; iNMB, incremental net monetary benefit; LMIC, low- and middle-income country; RR, rifampicin-resistant; SEAR, WHO Southeast Asian region; TB, tuberculosis; WHO, World Health Organization; YLD, years lost due to disability.

a 1× per-capita gross domestic product (GDP) cost-effectiveness threshold (both varied in sensitivity analyses). Vaccine introduction was estimated to require substantial near-term resources, offset by future cost-savings from averted TB burden. From a health system perspective, adolescent/adult vaccination was cost-effective in 64 of 105 LMICs. From a societal perspective (including productivity gains and averted patient costs), adolescent/adult vaccination was projected to be cost-effective in 73 of 105 LMICs and cost-saving in 58 of 105 LMICs, including 96% of countries with higher TB burden. When considering the monetized value of health gains, we estimated that introduction of an adolescent/adult vaccine could produce $283 to 474 billion in economic benefits by 2050. Limited data availability required assumptions and extrapolations that may omit important country-level heterogeneity in epidemiology and costs.

## Conclusions

TB vaccination would be highly impactful and cost-effective in most LMICs. Further efforts are needed for future development, adoption, and implementation of novel TB vaccines.

## Author summary

### Why was this study done?

- Previous studies have highlighted the economic impact of tuberculosis (TB) and the potential economic impact that novel TB vaccines could have on reducing this burden in specific low- and middle-income countries (LMICs).

- The cost and cost-effectiveness of novel TB vaccines, which depend on vaccine price and delivery strategy that may vary by country, are needed by vaccine developers, manufacturers, and potential purchasers to guide investment decisions.

- No modeling studies have estimated the cost and cost-effectiveness of novel TB vaccine products with country-specific assumptions for medical and non-medical costs, indirect costs, vaccine delivery costs, and delivery strategies across a wide range of LMICs.

### What did the researchers do and find?

- We estimated the costs, cost-effectiveness, and incremental net monetary benefit (iNMB) of TB vaccine introduction from both the health system and societal perspective, in order to inform global-level decision-making for novel TB vaccine investment and introduction.

- Using mathematical and economic models, we assessed scenarios for the introduction of novel TB vaccines with a wide range of characteristics and a diverse set of health and economic outcomes, including country-specific introduction years from 2028 to 2047.

- Our analysis projected that an effective new TB vaccine could offer large potential health and economic benefits over 2028 to 2050. From a societal perspective, vaccination was

projected to be cost-effective in 73 LMICs compared to a 1× per-capita gross domestic product (GDP) threshold.

- When considering the monetized value of health gains, we estimated that introduction of an adolescent/adult TB vaccine could produce $283 to 474 billion in health and economic benefits by 2050, with greater benefits in LMICs with elevated TB incidence.

### What do these findings mean?

- Introduction of a new TB vaccine was found to be impactful and cost-effective for a range of assumptions on vaccine price and delivery strategies, with aggregate health and economic benefits of similar scale to the most influential health interventions in LMIC settings in recent years.

- The results of these analyses can be used by global and country stakeholders to inform TB vaccine policy and introduction preparedness, as well as decision-making around future development, adoption, and implementation of novel TB vaccines.

## Introduction

Tuberculosis (TB) is one of the world's leading causes of infectious disease death [1]. It remains the leading cause of death for people living with HIV and a major contributor to anti-microbial-resistance-related deaths. The Coronavirus Disease 2019 (COVID-19) pandemic has reversed years of progress in providing TB services and, consequently, the number of people who died from TB increased to 1.5 million in 2020 [1].

The World Health Organization (WHO)'s End TB Strategy targets a 90% reduction in TB mortality and 80% decline in TB incidence by 2030, compared to 2015 [2]. Achieving these targets will require a comprehensive multisectoral response, along with transformational new tools. The cost of not meeting the End TB Targets by 2030 and facing the excess deaths resulting from COVID-19-related disruptions to TB services may translate into 31.8 million TB deaths globally corresponding to an economic loss of $17.5 trillion between 2020 and 2050 [3]. Developing new safe, affordable, and effective TB vaccines is critical for achieving these targets. While promising candidates exist (for example, the M72/AS01$_E$ candidate vaccine [4]), limited market incentives to invest in TB prevention has delayed the development of novel TB vaccines.

The WHO promotes the Full Value of Vaccines Assessment framework to improve decision-making around vaccine development and use [5,6]. As part of a Full Value of Vaccines Assessment of novel TB vaccines, we estimated the costs, cost-effectiveness, and net monetary benefit of TB vaccine introduction, from health system and societal perspectives, to inform global-level decision-making for novel TB vaccine investment and introduction [5,6].

## Methods

### Analytic overview

We estimated a range of outcomes quantifying the health and economic impact of new vaccine introduction for affected countries. To do so, we used linked epidemiological and economic

models to project changes in healthcare utilization, health outcomes, and healthcare costs for various vaccine introduction scenarios compared to a "no-new-vaccine" counterfactual. (Full epidemiological model details have been previously reported by Clark and colleagues [7] and are briefly described in Exhibit A in S1 Appendix. Any changes to the analysis that were required are also described; no prospective analysis plan was developed.) We estimated outcomes for each of 105 low- and middle-income countries (LMICs) over a 2028 to 2050 evaluation period (Exhibit B in S1 Appendix). We summarized results as the incremental costs, cost-effectiveness, and incremental net monetary benefits (iNMBs) of vaccine introduction. Results are presented for a range of analytic assumptions and introduction scenarios, organized by major country groupings (WHO region, World Bank income level [8], and WHO high-TB burden grouping [9]).

## Vaccination scenarios

We constructed a "no-new-vaccine" baseline with current TB interventions continuing into the future at current levels. Compared to this baseline, we evaluated 2 different vaccine product profiles: an infant "pre-infection" prevention of disease vaccine (i.e., efficacious for individuals uninfected at time of vaccination) with 80% efficacy targeting neonates and an adolescent/adult "pre- and post-infection" prevention of disease vaccine (i.e., efficacious in all individuals aside from those with active TB at time of vaccination) with 50% efficacy, based on WHO Preferred Product Characteristics for New Tuberculosis Vaccines [10]. For both vaccine product profiles, we assumed an average 10-year duration of protection, with exponential waning. We assumed the infant vaccine would be delivered through the routine vaccination program, with vaccine delivery at fixed sites following a standard dosing schedule. We assumed the adolescent/adult vaccine would be delivered through routine vaccination of 9-year-olds plus a one-time vaccination campaign for ages 10+. In the base-case scenario, we assumed countries would achieve linear scale-up to a specified coverage over 5 years. Based on consultation with global stakeholders, we assumed a coverage target of 85% for the infant vaccine (average coverage of diphtheria-tetanus-pertussis third dose for LMICs [11]), 80% for routine delivery of adolescent/adult vaccine, and 70% of the adolescent/adult vaccination campaign [12]. We selected 2028 as the earliest country-specific introduction year to align with anticipated product availability following TB vaccine candidate trial completion. We assumed country-specific introduction years from 2028 to 2047, determined based on indicators for disease burden, immunization capacity, classification of the country as an "early adopter/leader," lack of regulatory barriers, and commercial prioritization [7]. Further details regarding vaccination scenarios are provided in the appendix.

## Epidemiological outcomes and health service utilization

We projected future TB epidemiology and health service utilization using an age-structured TB transmission model calibrated to reported demography, TB burden estimates, and TB service utilization in each modeled country [7]. Out of 135 LMICs [8], we excluded 20 due to lack of critical calibration data and 10 due to unsuccessful calibration results (details provided in Exhibit A in S1 Appendix). We analyzed the remaining 105 countries (Exhibit B in S1 Appendix), representing 93.3% of global TB burden [13]. In countries in which the proportion of TB cases among people living with HIV was greater than or equal to 15%, and the HIV prevalence in the country was greater than 1%, the model included the effects of HIV and antiretroviral therapy (ART) on TB infection and progression risks (Exhibit A in S1 Appendix). Using this model, we estimated changes in TB epidemiology and related service utilization for each modeled scenario.

## Summary health outcomes

We estimated disability-adjusted life-years (DALYs) averted to quantify the health gains achieved by vaccination. To calculate years lost due to disability (YLDs), we assigned each modeled health state a disability weight from the Global Burden of Disease classification system (Exhibit C in S1 Appendix) [14]. For each scenario and year, total YLDs were calculated by summing life-years lived across all health states, weighted by the disability weight for each state. For each scenario and year, years of life lost were calculated by multiplying deaths at each year of age by reference life expectancy at that age [15] and summing across all ages.

## Cost outcomes

We estimated the costs of vaccine introduction, as well as changes in the costs of other health services (TB care, HIV care) by multiplying health service volume indicators (vaccines delivered, TB cases diagnosed and treated, ART patient-years) by country-specific unit costs. Diagnostics costs for drug-susceptible (DS) and rifampicin-resistant (RR) TB were obtained from published literature as average values for each country income level [16], which were assigned to each LMIC in the associated country income level grouping [8]. Unit costs for TB treatment were calculated as an average of country-level DS-TB [17] and income-level RR-TB [16,18] treatment costs, weighted by country-level RR-TB prevalence [1].

For ART costs, direct non-medical costs (travel, accommodation, food, nutritional supplements) to the patient, and productivity costs (income loss experienced by patients during TB care), we derived unit costs by extrapolating estimates reported by the Global Health Cost Consortium [19] (sample size = 39) and WHO patient cost surveys (sample size = 20) [20,21] with meta-regression models for the respective outcomes specified as generalized linear regression models (GLMs), assuming a Gamma distributed outcome, a log link function, and gross domestic product (GDP) per capita as the predictor [22]. The previous unit costs were inflated to 2020 values in local currency using the country consumer price index [23] and converted to 2020 US dollars using market exchange rates [24].

Productivity costs due to premature death were estimated as the incremental number of life-years gained under a given vaccination scenario, multiplied by 2020 per-capita GDP as an approximation of income.

As the per-dose cost for novel TB vaccines is unclear while products are still under development, the base-case used an LMIC price of human papillomavirus (HPV) vaccine ($4.60) for a novel vaccine proxy with an injection supply cost per dose of $0.11 and 5% wastage [25,26]. Country-specific vaccine delivery costs were based on a meta-analysis of childhood [27] and HPV vaccine delivery unit costs for the infant and adolescent/adult vaccines, respectively, plus additional one-time vaccine introduction costs ($0.65 and $2.40 per targeted individual in the first year of introduction for infant and adolescent/adult vaccines, respectively) [28]. Costs are reported in 2020 US dollars.

## Cost-effectiveness analysis

Incremental cost-effectiveness ratios (ICERs) were calculated from health system and societal perspectives, with a 3% discount rate, across the 2028 to 2050 evaluation period. We also reported a specification in which costs are discounted but not health outcomes. The health system perspective considered costs of vaccine introduction, plus the costs of TB and HIV services indirectly affected by vaccine introduction. The societal perspective additionally included patient non-medical and productivity costs. ICERs were compared to a range of country-specific cost-effectiveness thresholds to reflect the lack of consensus for a single threshold, including multiples of per-capita GDP [29] (assuming 1× per-capita GDP as a proxy for willingness

to pay in the base-case), recent estimates of the opportunity cost of healthcare spending [30,31], and the WHO's universal "Best Buy" threshold of $100 per DALY averted. This study is reported as per the Consolidated Health Economic Evaluation Reporting Standards 2022 (CHEERS 2022) statement (Exhibit D in S1 Appendix) [32].

### Return on investment

We quantified the return on investment as the iNMB from the societal perspective of each vaccine scenario compared to baseline for a range of willingness-to-pay thresholds [29–31]. iNMB was calculated as the sum of monetized health gains (DALYs averted multiplied by the estimated willingness-to-pay per DALY averted) minus incremental costs. We estimated the market size for each vaccine product profile, summing all individuals across 2028 to 2050 who were vaccinated in the model in the base-case scenario in countries in which the vaccine was cost-effective (ICER less than per-capita GDP). We also estimated market size based on countries in which vaccination was cost-saving under the societal perspective.

### Statistical analysis

We explored estimation uncertainty using a second-order Monte Carlo simulation [33]. We constructed probability distributions representing uncertainty in economic inputs and disability weights, specified as Gamma distributions for parameters defined over $[0, \infty]$, and Beta distributions for parameters defined over $[0, 1]$. For each parameter, the distribution mean was set equal to the point estimate, and the dispersion was set so an equal-tailed 95% interval reproduced the reported interval width. For parameters estimated from a meta-regression model (ART costs, patient costs), we simulated parameter values from each fitted regression model. We drew 1,000 random values for each uncertain parameter. We represented uncertainty in healthcare utilization and epidemiological outcomes (counts of each outcome by scenario, year, and population stratum) using 1,000 results sets from the transmission-dynamic model. This analysis generated 1,000 estimates for each outcome of interest, which we summarized as equal-tailed 95% uncertainty intervals.

### Sensitivity analysis

Compared to the base-case coverage targets (85%, 80%, 70% for routine infant vaccine delivery, routine adolescent vaccine delivery, and campaign adolescent/adult vaccine delivery, respectively), we examined a low-coverage scenario (75%, 70%, and 50%, respectively) and a high-coverage scenario (95%, 90%, and 90%, respectively).

We examined 2 alternative vaccine delivery scenarios. First, we modeled an accelerated scale-up scenario in which all countries introduced vaccination in 2025 and achieved instantaneous scale-up to the specified coverage target. Second, we modeled a routine-only scenario that removed the one-time campaign-delivery component of the adolescent/adult base-case scenario.

We examined 3 alternative vaccine price scenarios, including scenarios in which the base-case vaccine price of $4.60 was both halved ($2.30) and doubled ($9.20), respectively. A third scenario examined high-middle-tier vaccine pricing, with higher prices for middle-income countries based on UNICEF vaccine pricing data ($10.25 for non-Gavi countries with gross national income (GNI) per capita less than $3,995 and $14.14 for non-Gavi countries with GNI per capita greater than $3,995; Exhibit B in S1 Appendix) [25].

Compared to the base-case scenario assuming 1 vaccine dose, we estimated results assuming that 2 vaccine doses were required to achieve the same level of efficacy, i.e., a full vaccination course required 2× the base-case vaccine price of $4.60 and 2× the delivery cost.

We also estimated results with an alternative set of assumptions about TB incidence trends in the no-new-vaccine baseline, with incidence assumed to decline more rapidly through the scale-up of existing preventive treatment and case detection, meeting the 2025 "End TB" incidence reduction target without introduction of a new vaccine [2].

Compared to the base-case assumption of 10-year duration of protection, we also examined lifelong duration of protection conferred by vaccination.

Finally, compared to the base-case assumption of 50% efficacy for the adolescent/adult vaccine, we also examined 75% efficacy conferred by this vaccine.

## Results

### Costs and cost-effectiveness analysis

A summary of the unit costs by country income level is provided in Table 1. In the no-new-vaccine baseline, over 2028 to 2050, total undiscounted (i.e., without a 3% annual discount rate) costs of TB diagnosis and treatment were estimated to be $20.7 (95% uncertainty interval: 12.8 to 31.2) billion for DS-TB and $19.2 (15.6 to 23.1) billion for RR-TB (Exhibit E in S1 Appendix). For the infant vaccine scenario, vaccine introduction costs were $11.8 (9.6 to 16.9) billion, and averted TB diagnosis and treatment costs were $342 (223 to 489) million for DS-TB and $299 (251 to 351) million for RR-TB over 2028 to 2050 (Exhibit F in S1 Appendix). For the adolescent/adult vaccine scenario, vaccine introduction costs were $50.5 (38.1 to 75.9) billion, and averted TB diagnosis and treatment costs were $3.5 (2.2 to 5.2) billion for DS-TB and $3.2 (2.6 to 3.8) billion for RR-TB over 2028 to 2050 (Exhibit G in S1 Appendix)—greater than the averted disease costs in the infant vaccine scenario. There would also be $13.4 (9.5 to 19.2) million and $362 (281 to 466) million in additional ART costs under the infant and adolescent/adult vaccine scenarios, respectively, due to extended survival among people living with HIV (Exhibits H and I in S1 Appendix).

There was greater, and more rapid, impact from an adolescent/adult vaccine compared to an infant vaccine over the study period (Exhibits J and K in S1 Appendix). Across 2028 to 2050, infant vaccine costs were projected to increase smoothly from the year of vaccine introduction, whereas the adolescent/adult vaccine scenario required major upfront investments during vaccine introduction and 5-year campaign roll-out, then decreased substantially after campaigns were completed.

In the base-case analysis, from the health system perspective, we found that infant vaccination would be cost-effective (ICER below 1-times per-capita GDP) compared to no vaccination in 47 of 105 modeled LMICs (45%) and 24 of 27 with high-TB burden (89%). Using the same assumptions, we found that adolescent/adult vaccination would be cost-effective in 64 out of 105 countries (61%) and all 27 with high-TB burden. Neither vaccine strategy would be cost-saving in any country. Fig 1 displays the distribution of country-level cost-effectiveness results from the health system perspective for infant and adolescent/adult vaccines, stratified by TB incidence level. Vaccine introduction was more likely to be cost-effective in countries with higher TB incidence.

From the societal perspective, the infant vaccine was cost-effective in 56 out of 105 countries (53%), including all with high-TB burden, and cost-saving in 46 countries (44%). Similarly, the adolescent/adult vaccine was cost-effective in 73 out of 105 countries (70%), remaining cost-effective in all with high-TB burden, and cost-saving in 58 countries (55%). Fig 2 displays the percentage of the modeled population that live in countries where vaccination was cost-effective based on different cost-effectiveness thresholds (Exhibit L in S1 Appendix shows the percentage of countries where vaccination was cost-effective; Exhibits M and N in S1 Appendix present tabular results).

**Table 1. Unit cost inputs by country income level.**

|  | LIC | LMIC | UMIC |
|---|---|---|---|
| Drug-susceptible TB diagnostics costs | 21.67 (17.55, 26.47) | 50.27 (40.71, 61.42) | 66.87 (54.15, 81.7) |
| Rifampicin-resistant TB diagnostics costs | 430.16 (349.13, 524.53) | 404.51 (327.57, 494.20) | 343.73 (278.35, 419.95) |
| Drug-susceptible TB treatment costs | 708.00 (445.68, 1051.44) | 1,087.80 (640.44, 1,693.08) | 1,812.24 (1,023.36, 2,884.08) |
| Rifampicin-resistant TB treatment costs | 1,538.09 (1309.96, 1798.54) | 3,631.83 (3,269.94, 4,033.66) | 3,974.10 (3,593.33, 4,395.81) |
| ART costs | 210.47 (204.30, 216.70) | 321.08 (310.26, 331.93) | 509.96 (488.81, 532.07) |
| Direct non-medical costs for drug-susceptible TB | 343.17 (327.55, 359.71) | 343.90 (327.08, 361.52) | 500.59 (468.36, 535.26) |
| Income loss experienced by patients during drug-susceptible TB care | 343.48 (327.22, 360.71) | 342.85 (325.97, 360.54) | 500.59 (468.36, 535.26) |
| Direct non-medical costs for rifampicin-resistant TB | 1,373.14 (1,293.92, 1,459.45) | 1,397.84 (1,309.87, 1,492.72) | 2,240.79 (2,057.04, 2,450.43) |
| Income loss experienced by patients during rifampicin-resistant TB care | 1,380.06 (1281.06, 1466.81) | 1,393.65 (1,305.31, 1,488.93) | 2,240.79 (2,057.04, 2,450.43) |
| Infant vaccine delivery costs | 1.61 (0.51, 3.96) | 2.89 (1.08, 6.51) | 4.61 (1.78, 10.64) |
| Adolescent/adult vaccine delivery costs | 3.05 (1.50, 5.68) | 3.25 (1.62, 6.27) | 3.80 (1.24, 9.50) |

Values in parentheses represent equal-tailed 95% credible intervals. LIC: GNI per capita of $1,085 or less; LMIC: GNI per capita of $1,086 to $4,225; UMIC: GNI per capita of $4,256 to $13,205 (World Bank 2021).

ART, antiretroviral therapy; GNI, gross national income; LIC, low-income country; LMIC, lower middle-income country; TB, tuberculosis; UMIC, upper middle-income country.

Tables 2 and 3 report summary health outcomes, costs, and cost-effectiveness of the base-case vaccination scenarios. Across all 105 analyzed countries, the majority of TB cost-savings accrued in high-TB-burden settings, particularly in lower middle-income settings and WHO African region (AFR) and Southeast Asian region (SEAR). Assuming 0% discounting on health outcomes decreased ICERs (indicating greater cost-effectiveness) for the infant vaccine by approximately 76% and for the adolescent/adult vaccine by approximately 69% from the health system perspective (Exhibits O and P in S1 Appendix).

## Return on investment

With each averted DALY valued at per-capita GDP and costs assessed from the societal perspective, we estimated a cumulative $68.6 (range: $44.5 to 100 across examined thresholds) billion iNMB globally for infant vaccine introduction in countries where introduction was cost-effective at 1-times per-capita GDP (Fig 3; tabular results in Exhibit Q in S1 Appendix). For the adolescent/adult vaccine, we estimated iNMB of $372 billion for countries in which vaccination was cost-effective (range: $283 to 474 billion). These benefits were concentrated in regions (AFR, SEAR) with higher disease burden. For the infant vaccine, the market size (i.e., the vaccinated population in countries in which the vaccine would be cost-effective at per-capita GDP from the societal perspective) would be 1.431 (1.430 to 1.432) billion individuals, while for the adolescent/adult vaccine, this population size would be 5.182 (5.180 to 5.183) billion individuals. Under a more restrictive assumption where the vaccine is only introduced in countries where the societal ICER is cost-saving, the market size would be 1.316 (1.315 to 1.317) billion individuals for the infant vaccine, and 4.642 (4.617 to 4.644) billion individuals for the adolescent/adult vaccine. The largest markets were in the WHO SEAR and WPR regions (Exhibits R and S in S1 Appendix).

## Sensitivity analysis

From both health system and societal perspectives, DALYs averted and costs decreased in the low-coverage scenario, and increased in the high-coverage scenario, for both the infant and

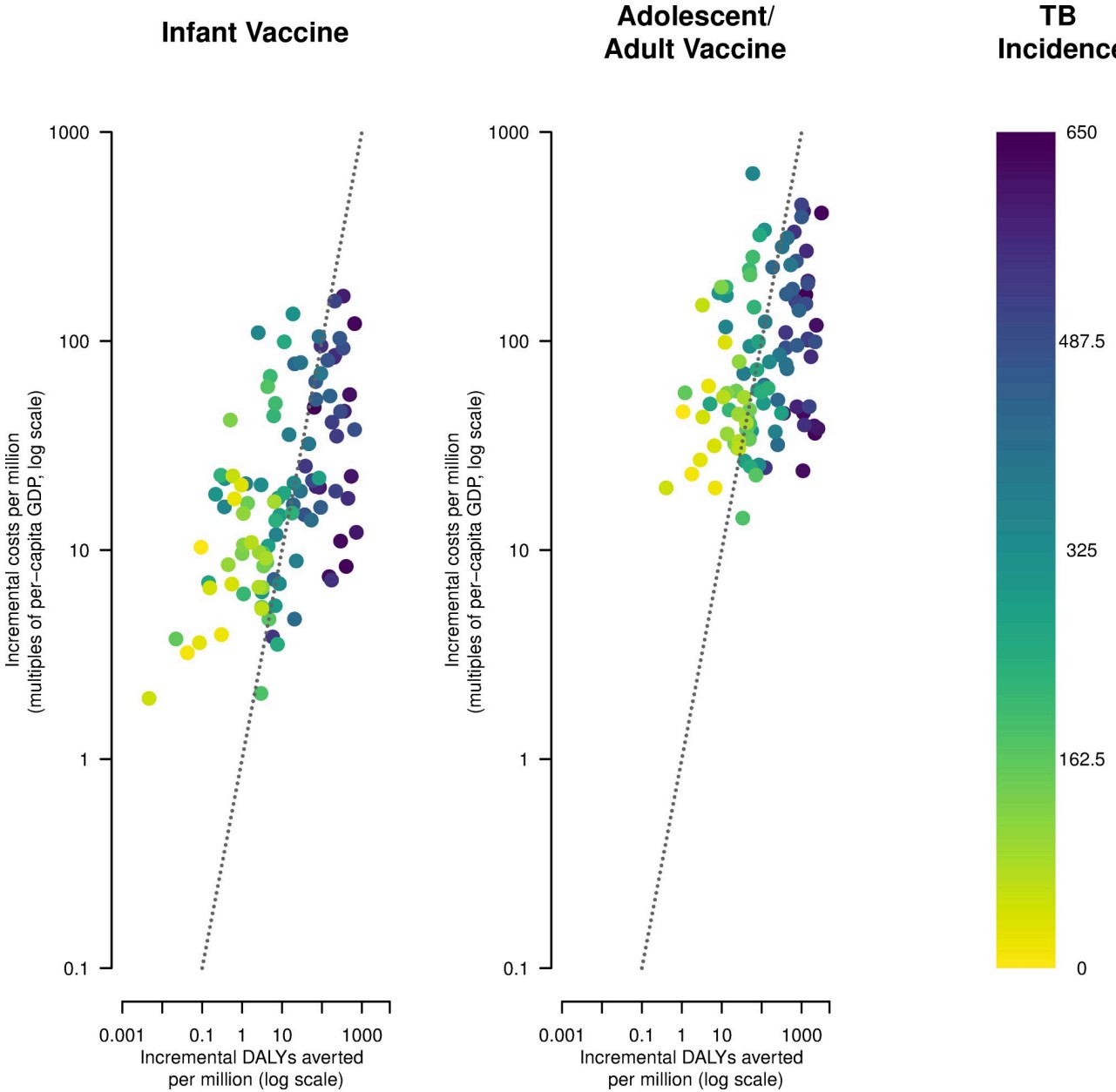

**Fig 1. Cost-effectiveness results from the health system perspective for novel tuberculosis vaccines by country and vaccine.** Note: Points represent each of 105 LMICs analyzed in the base-case scenario, stratified by TB incidence per 100,000. Line represents a cost-effectiveness threshold of 1× per-capita GDP in 2020. Vaccine introduction would be considered cost-effective for countries falling underneath this line. DALY, disability-adjusted life year; GDP, gross domestic product; LMIC, low- and middle-income country; TB, tuberculosis.

the adolescent/adult vaccine, with evidence of diminishing returns as coverage increases (Exhibits T and W in S1 Appendix).

Compared to the base-case vaccination introduction and delivery scenario, the accelerated scale-up scenario had greater health impact (DALYs averted) and better cost-effectiveness (assuming per-unit vaccination costs were unchanged), with vaccination being cost-effective in all 105 LMICs for both the infant and adolescent/adult vaccine compared to a per-capita GDP threshold (Exhibits X–AA in S1 Appendix). Conversely, the routine-only scenario had a

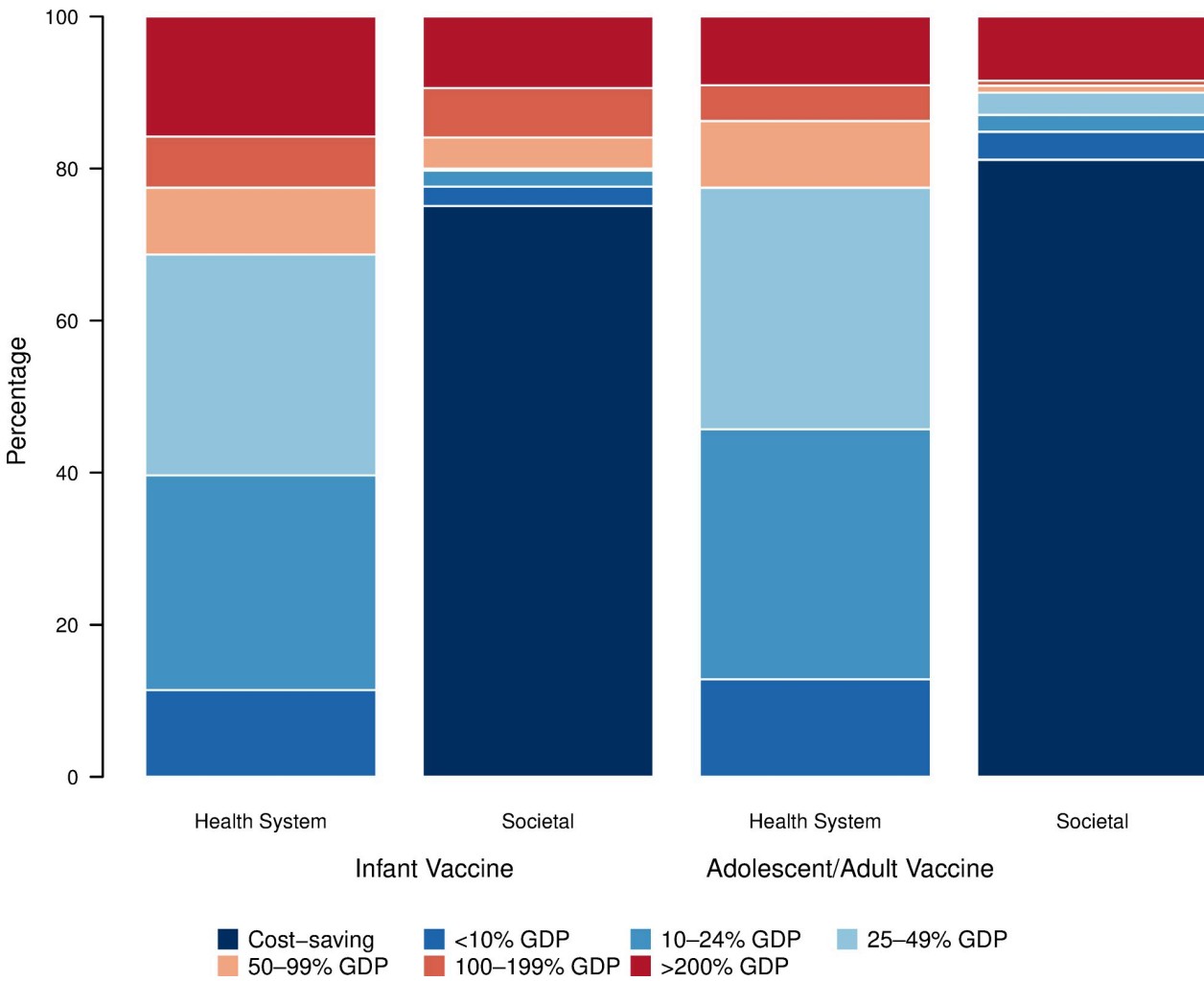

**Fig 2. Percentage of population that lives in countries where vaccination was cost-effective compared to percentage of GDP per capita thresholds, comparing health system and societal perspectives.** Note: Countries include 105 LMICs analyzed. Population includes vaccinated individuals 2028–2050. GDP per capita estimates from 2020. GDP, gross domestic product per capita; LMIC, low- and middle-income country.

much smaller health impact and modestly worse cost-effectiveness profile, as compared to the base-case analysis (Exhibit BB in S1 Appendix).

Reducing the vaccine price by half decreased infant vaccination costs from $11.8 to $7.6 billion (36% decrease) and adolescent/adult vaccination costs from $50.5 to $36.4 billion (28% decrease; Exhibits CC–LL in S1 Appendix). Doubling the vaccine price increased infant vaccination costs to $20.2 billion (71% increase) and adolescent/adult vaccination costs to $78.8 billion (56% increase). Switching to high-middle-tier vaccine pricing (higher vaccine prices for middle-income countries) increased infant vaccine costs to $16.9 billion (44% increase) and adolescent/adult vaccination costs to $72.1 billion (43% increase). From the health system perspective, reducing the vaccine price by half increased the number of countries in which infant vaccination was cost-effective at a per-capita GDP threshold from 47 to 51, whereas doubling the vaccine price decreased the number of cost-effective countries to 32. Assuming higher vaccine prices for middle-income countries reduced the number of countries in which the infant vaccine was cost-effective at a per-capita GDP threshold from 47 to 41. Similarly, the half-price scenario, double-price scenario, and high-middle-tier-price scenario changed the number of

**Table 2. Discounted costs, DALYs averted, and cost-effectiveness of infant tuberculosis vaccines.**

| Country grouping | Health system perspective[a] incremental cost (USD billions) | Societal perspective[b] incremental cost (USD billions) | DALYs averted (millions) | Health system cost (USD) per DALY averted | Societal cost (USD) per DALY averted |
|---|---|---|---|---|---|
| All countries | 7.67 (6.25, 10.6) | −27.4 (−32.6, −22.3) | 18.0 (15.6, 20.8) | 428 (331, 606) | cost-saving[c] |
| High-TB burden[d] | 5.15 (4.26, 7.14) | −28.1 (−33.1, −23.3) | 16.6 (14.2, 19.5) | 311 (241, 443) | cost-saving[c] |
| High-TB/HIV burden[d] | 3.62 (3.03, 4.88) | −24.9 (−29.9, −20.5) | 14.5 (12.2, 17.3) | 251 (194, 350) | cost-saving[c] |
| High-MDR/RR-TB burden[d] | 4.53 (3.72, 6.33) | −26.5 (−31.4, −21.7) | 14.7 (12.4, 17.4) | 311 (236, 449) | cost-saving[c] |
| **Income level[e]** | | | | | |
| LIC | 1.12 (0.96, 1.44) | −0.19 (−0.48, 0.15) | 2.02 (1.70, 2.46) | 558 (431, 737) | cost-saving (cost-saving, 77.2) |
| LMIC | 3.95 (3.31, 5.31) | −23.1 (−27.8, −18.8) | 14.9 (12.6, 17.6) | 268 (208, 374) | cost-saving[c] |
| UMIC | 2.60 (1.96, 4.12) | −4.15 (−6.24, −2.25) | 1.12 (0.85, 1.50) | 2,380 (1,490, 3,870) | cost-saving[c] |
| **World region** | | | | | |
| AFR | 2.40 (2.05, 3.13) | −13.4 (−17.1, −10.4) | 9.34 (7.72, 11.3) | 260 (196, 352) | cost-saving[c] |
| AMR | 0.69 (0.52, 1.07) | 0.29 (0.10, 0.67) | 0.06 (0.05, 0.08) | 10,900 (7,680, 16,900) | 4,630 (1,400, 10,700) |
| EMR | 1.05 (0.86, 1.42) | −0.62 (−1.32, −0.01) | 1.64 (1.15, 2.25) | 661 (430, 1,010) | cost-saving[c] |
| EUR | 0.43 (0.30, 0.72) | 0.20 (0.07, 0.49) | 0.04 (0.03, 0.04) | 11,100 (7,530, 18,900) | 5260 (1690, 12900) |
| SEAR | 1.59 (1.31, 2.18) | −9.90 (−12.9, −7.37) | 5.50 (4.15, 7.31) | 296 (204, 450) | cost-saving[c] |
| WPR | 1.50 (1.16, 2.33) | −4.03 (−5.74, −2.62) | 1.42 (1.04, 1.95) | 1,080 (688, 1,770) | cost-saving[c] |

[a] Costs from the health system perspective include vaccination costs, tuberculosis testing and treatment costs, and antiretroviral treatment costs.

[b] Costs from the societal perspective include health system perspective costs, as well as patient non-medical costs and productivity losses.

[c] Both the point estimate and the interval estimates are cost-saving.

[d] High-TB, high-TB/HIV (HIV-associated TB), and high-MDR/RR-TB (multidrug/rifampicin-resistant TB) burden countries as defined by the World Health Organization [9].

[e] LIC: GNI per capita of $1,085 or less; LMIC: GNI per capita of $1,086 to $4,225; UMIC: GNI per capita of $4,256 to $13,205 (World Bank 2021).

All countries include 105 LMICs analyzed. Values in parentheses represent equal-tailed 95% credible intervals.

AFR, African region; AMR, Region of the Americas; DALY, disability-adjusted life-year; EMR, Eastern Mediterranean region; EUR, European region; GDP, gross domestic product; GNI, gross national income; LIC, low-income; LMIC, lower middle-income country; SEAR, Southeast Asian region; UMIC, upper middle-income; USD, United States dollar; WPR, Western Pacific region.

countries in which the adolescent/adult vaccine was considered cost-effective from the health system perspective at a per-capita GDP threshold from 64 in the base-case, to 70, 52, and 55, respectively.

Assuming a two-dose vaccination course increased infant vaccination costs to $23.3 billion (Exhibit CC in S1 Appendix) and adolescent/adult vaccination costs to $100 billion (Exhibit DD in S1 Appendix). From the health system perspective, a two-dose vaccination course decreased the number of countries in which infant vaccination was cost-effective at a per-capita GDP threshold from 47 to 31 and in which adolescent/adult vaccination was cost-effective from 64 to 46 (Exhibits MM and NN in S1 Appendix).

Assuming the no-new-vaccine baseline with faster incidence reductions through strengthening of current TB interventions to meet the 2025 End TB targets, a number of countries remained cost-saving from the societal perspective (9 countries for infant vaccine and 20 countries for adolescent/adult vaccine; Exhibits OO and PP in S1 Appendix).

An infant vaccine with lifelong duration of protection averted 30.0 (26.1 to 34.5) million DALYs, a 66% increase compared to the base-case assumption of 10-year protection (Exhibit QQ in S1 Appendix). An adolescent/adult vaccine with lifelong duration of protection averted 138 (127 to 150) million DALYs (46% greater than the base-case; Exhibit RR in S1 Appendix).

**Table 3. Discounted costs, DALYs averted, and cost-effectiveness of adolescent/adult tuberculosis vaccines.**

| Country grouping | Health system perspective[a] incremental cost (USD billions) | Societal perspective[b] incremental cost (USD billions) | Incremental DALYs averted (millions) | Health system cost (USD) per DALY averted | Societal cost (USD) per DALY averted |
|---|---|---|---|---|---|
| All countries | 36.3 (26.8, 55.2) | −145 (−165, −123) | 94.8 (86.9, 103) | 383 (276, 579) | cost-saving[c] |
| High-TB burden[d] | 26.0 (19.0, 39.8) | −142 (−159, −122) | 86.5 (78.5, 95.1) | 301 (213, 462) | cost-saving[c] |
| High-TB/HIV burden[d] | 15.8 (11.8, 22.9) | −131 (−147, −115) | 77.1 (69.6, 85.3) | 205 (148, 299) | cost-saving[c] |
| High-MDR/RR-TB burden[d] | 24.0 (17.2, 37.5) | −133 (−151, −114) | 76.9 (69.2, 85.3) | 312 (217, 486) | cost-saving[c] |
| Income level[e] | | | | | |
| LIC | 3.73 (2.98, 4.98) | −2.57 (−3.70, −1.20) | 9.53 (8.40, 10.7) | 394 (301, 535) | cost-saving[c] |
| LMIC | 17.1 (13.0, 24.3) | −119 (−133, −105) | 78.2 (70.8, 86.3) | 219 (160, 317) | cost-saving[c] |
| UMIC | 15.5 (10.1, 27.4) | −24.0 (−33.8, −9.93) | 7.15 (5.89, 8.83) | 2,190 (1,320, 4,140) | cost-saving[c] |
| World region | | | | | |
| AFR | 6.89 (5.51, 9.26) | −56.6 (−66.4, −48.3) | 38.9 (35.0, 42.8) | 178 (137, 246) | cost-saving[c] |
| AMR | 3.84 (2.60, 6.56) | −0.17 (−1.54, 2.67) | 0.73 (0.65, 0.81) | 5,300 (3,510, 9,050) | cost-saving (cost-saving, 3,590) |
| EMR | 4.06 (3.19, 5.61) | −2.65 (−4.45, −0.70) | 6.61 (5.22, 8.24) | 623 (438, 908) | cost-saving[c] |
| EUR | 2.08 (1.40, 3.56) | −0.64 (−1.41, 0.82) | 0.50 (0.45, 0.56) | 4,130 (2,730, 7,260) | cost-saving (cost-saving, 1710) |
| SEAR | 9.38 (7.01, 13.6) | −71.3 (−82.6, −59.8) | 41.7 (35.8, 48.3) | 226 (159, 338) | cost-saving[c] |
| WPR | 10.0 (6.55, 17.6) | −13.8 (−18.5, −5.79) | 6.37 (5.55, 7.31) | 1,580 (1,000, 2,810) | cost-saving[c] |

[a] Costs from the health system perspective include vaccination costs, tuberculosis testing and treatment costs, and antiretroviral treatment costs.

[b] Costs from the societal perspective include health system perspective costs, as well as patient non-medical costs and productivity losses.

[c] Both the point estimate and the interval estimates are cost-saving.

[d] High-TB, high-TB/HIV (HIV-associated TB), and high-MDR/RR-TB (multidrug/rifampicin-resistant TB) burden countries as defined by the World Health Organization [9].

[e] LIC: GNI per capita of $1,085 or less; LMIC: GNI per capita of $1,086 to $4,225; UMIC: GNI per capita of $4,256 to $13,205 (World Bank 2021).

All countries include 105 LMICs analyzed. Values in parentheses represent equal-tailed 95% credible intervals.

AFR, African region; AMR, Region of the Americas; DALY, disability-adjusted life-year; EMR, Eastern Mediterranean region; EUR, European region; GDP, gross domestic product; GNI, gross national income; LIC, low-income; LMIC, lower middle-income country; SEAR, Southeast Asian region; UMIC, upper middle-income; USD, United States dollar; WPR, Western Pacific region.

Both vaccine products remained cost-saving from the societal perspective, assuming lifelong duration of protection decreased the ICER by approximately 42% and 36% for the infant and adolescent/adult vaccine, respectively (health system perspective).

An adolescent/adult vaccine with 75% efficacy also averted 138 (127 to 150) million DALYs (45% greater than the base-case; Exhibit SS in S1 Appendix), but decreased the ICER to a greater degree than lifelong duration of protection at 53%.

## Discussion

An effective novel TB vaccine would offer large potential health and economic benefits over 2028 to 2050. The results of this analysis demonstrate that, when available, TB vaccines could be cost-effective in a majority of LMICs, particularly from the societal perspective, and in essentially all high-burden countries. Introducing novel TB vaccines could also offer high value in terms of iNMB to patients, the health system, and society, particularly in countries with high burden of TB, HIV-associated TB, and/or multidrug-resistant/RR-TB.

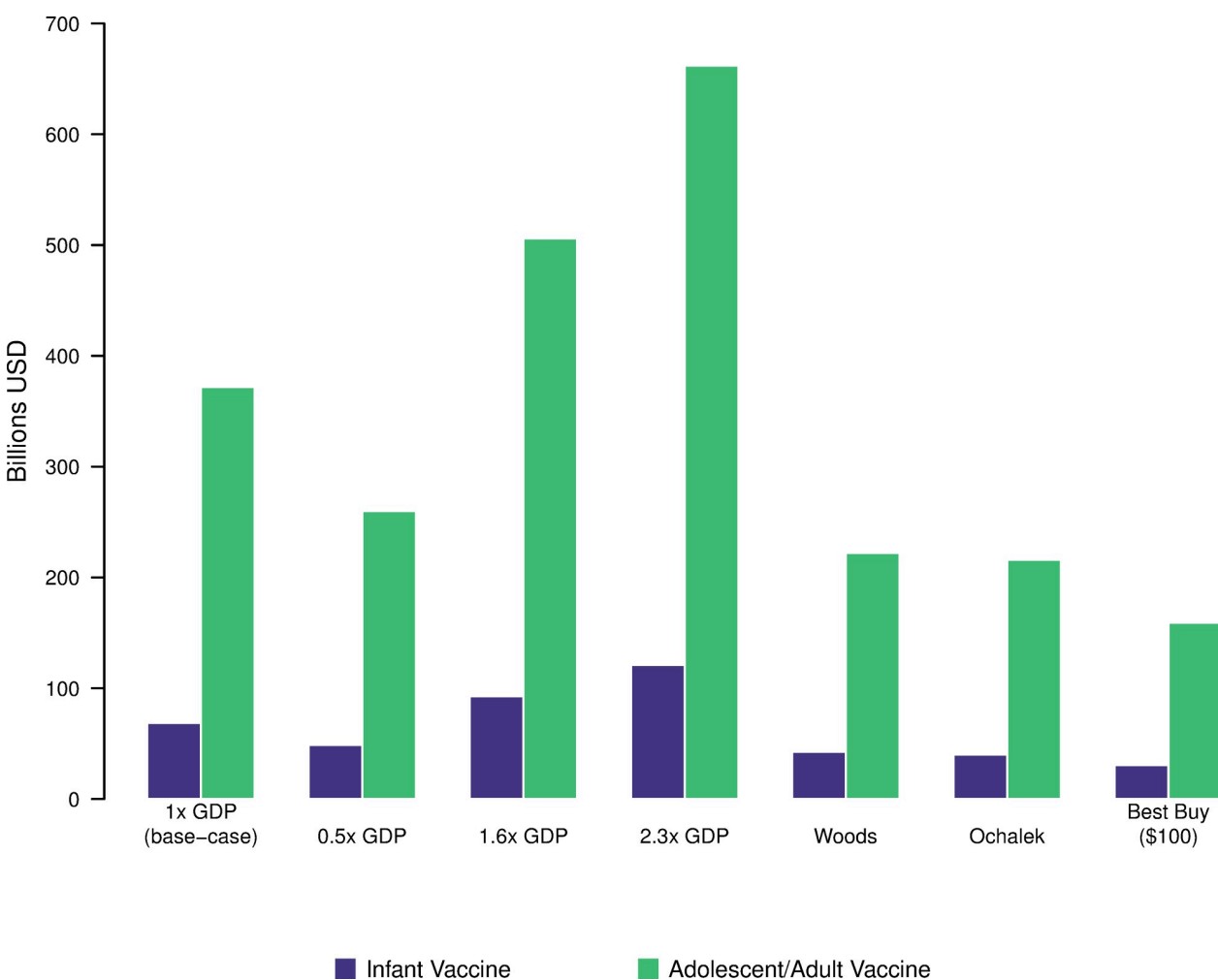

**Fig 3. Incremental net monetary benefit of novel tuberculosis vaccines assessed from the societal perspective, for several willingness-to-pay thresholds.** Note: Estimates include the iNMB from the countries that are cost-effective at the respective threshold [29–31]. GDP per capita estimates from 2020. GDP, gross domestic product per capita; iNMB, incremental net monetary benefit.

For both vaccine product profiles, vaccination was more likely to be cost-effective in lower middle-income countries (relative to low-income and upper middle-income countries), as countries in this income group are more likely to have both significant TB burden and sufficient economic resources to justify additional TB investments without displacing other important health interventions. Vaccination was more frequently cost-effective in AFR and SEAR regions and the adolescent/adult vaccine was estimated to be cost-effective in all countries in the 27 modeled high-TB burden countries that accounted for 81.8% of global incident TB cases and 80.9% of global TB deaths in 2020 [9,13]. This relationship of ICERs decreasing as disease burden increases has also been shown for several licensed vaccines, including HPV, malaria, and rotavirus [34–36]. Although vaccines can be economically less viable for manufacturers, we estimated large potential markets for vaccinees in high-burden, middle-income settings. We estimated cost-effectiveness using a range of cost-effectiveness benchmarks, based on recent discussion of the validity of conventional standards [31,37–39]. Final decisions about vaccine adoption will be made by local decision-makers, based on the values placed on

health benefits, opportunity costs, the relative timing of health outcomes and costs, and other context-specific considerations.

There was greater, and more rapid, impact from an adolescent/adult vaccine over the 2028 to 2050 time horizon compared to an infant vaccine, as this vaccine is targeted to a population with the highest burden of TB, and the delay between vaccination and TB prevention impact is shorter with the adolescent/adult vaccine. For the adolescent/adult vaccine, we estimated major short-term costs from introduction and one-time vaccination campaigns, with the highest costs incurred during the 10 years following vaccine introduction. In contrast, the cost-savings from averted TB cases were realized gradually over 2028 to 2050, growing in magnitude towards the end of the time horizon. By assuming the no-new-vaccine baseline meeting the End TB targets, the remaining TB burden that could be averted by vaccination was estimated to be smaller, yielding results that were less cost-effective.

There are several factors that distinguish this analysis from past studies [40–43]: firstly, the steps taken to construct a realistic vaccine adoption timeline, based on an analysis of factors affecting country adoption decisions and stakeholder consultation. As a result, this analysis provides a more robust estimate of the potential timing of vaccine impact compared to past analyses, which is particularly important given the expected role of vaccines in contributing to the attainment of TB elimination targets for specific calendar years. Secondly, this study examined a wide range of scenarios for vaccine introduction, illustrating how the pace and extent of scale-up affect overall impact by 2050. In particular, the comparison of the base-case scenario to an accelerated scale-up scenario demonstrates the additional health and economic gains that would be possible with more rapid vaccine introduction. Thirdly, this study estimated iNMB as a single measure summarizing the health and economic benefits of TB vaccine introduction. Combined with the large number of countries modeled, this analysis quantifies the overall global benefits of TB vaccination in a way that can be set against the costs and other challenges that must be overcome to develop a new vaccine. This is particularly important for justifying the investments that still need to be made in vaccine development and preparation for deployment.

This analysis had several limitations. We were constrained by data availability, with only 105 countries successfully parameterized and calibrated. However, these 105 countries represent 93.3% of LMIC TB incidence and 93.6% of LMIC TB mortality globally in 2020 [13]. As a "pre- and post-infection" vaccine, the adolescent/adult vaccine was assumed to be equally effective regardless of previous infection status, which may have led to an overestimation of averted TB cases and deaths if the vaccine is less effective in infected vaccinees. We also restricted the analysis to focus on vaccine products that would prevent development of active TB, and did not examine the possible impact of vaccine products that would prevent infection, such as recombinant BCG vaccines [44]. If successfully developed, such vaccines could provide another effective tool for accelerating TB control. We also extrapolated from published literature [16–18] for several major unit cost inputs, potentially omitting important country-level heterogeneity in these costs. The sample size of patient cost surveys used for non-medical and productivity costs was small (20); therefore, the extrapolation to other country settings may not capture the level of potential variation in these costs. We did not investigate targeting high-risk subgroups for vaccination; vaccination could still be cost-effective when targeted to subgroups in settings where vaccination was not estimated to be cost-effective in a national roll-out. In our main analysis, we used 1× per-capita GDP as a threshold to approximate willingness-to-pay per DALY averted, based on its historical broad utilization in economic evaluation studies. However, recent evidence suggests that a more stringent cost-effectiveness threshold may be appropriate [30,31], in which case vaccination would be cost-effective in fewer countries and estimates of return on investment would be lower. Finally, we did not

consider all possible product and introduction scenarios, such as varying ages of vaccination, vaccine coverage targets by country, and scale-up trends by country, but demonstrated the potential value of novel TB vaccines according to specified characteristics. Future work developing detailed country-level models could take into account the health system capacity of each country and the underlying country-specific TB epidemiology by age to inform more realistic delivery scenarios.

Across this analysis, introduction of a novel TB vaccine was found to be impactful and cost-effective for a range of assumptions on vaccine price and delivery strategies, with aggregate health and economic benefits of similar scale to the most influential health interventions in LMIC settings in recent years [45]. TB vaccines are still under development, so their potential effectiveness and impact are uncertain. Accelerating the timeline for vaccine introduction, decreasing the vaccine price, or increasing vaccine efficacy could all impact the cost-effectiveness profile of vaccination and increase the magnitude of the benefits, directly improving the welfare of individuals and households that would otherwise experience the health and economic consequences of TB in coming years. Future work should investigate country-level vaccine policy questions to support introduction preparedness. The results of these analyses can be used by global and country stakeholders to inform these questions, as well as decision-making around future development, adoption, and implementation of novel TB vaccines.

## Supporting information

**S1 Appendix.**
(DOCX)

## Acknowledgments

We thank all the attendees at the WHO meetings on the Full Value Assessment of TB Vaccines for insightful advice and direction. The views expressed are those of the authors and do not necessarily represent the views of their respective organizations.

### Ethics approval and consent to participate

Not applicable.

## Author Contributions

**Conceptualization:** Allison Portnoy, Matthew Quaife, Mark Jit, Richard G. White, Nicolas A. Menzies.

**Data curation:** Allison Portnoy, Rebecca A. Clark, Inés Garcia Baena, Nobuyuki Nishikiori.

**Formal analysis:** Allison Portnoy, Rebecca A. Clark, Matthew Quaife, Chathika K. Weerasuriya, Christinah Mukandavire, Roel Bakker, Arminder K. Deol, Mark Jit, Richard G. White, Nicolas A. Menzies.

**Funding acquisition:** Richard G. White, Nicolas A. Menzies.

**Methodology:** Allison Portnoy, Matthew Quaife, Shelly Malhotra, So Yoon Sim, Raymond C. W. Hutubessy, Inés Garcia Baena, Mark Jit, Richard G. White, Nicolas A. Menzies.

**Resources:** Nebiat Gebreselassie, Matteo Zignol.

**Supervision:** Richard G. White, Nicolas A. Menzies.

**Writing – original draft:** Allison Portnoy, Rebecca A. Clark, Richard G. White, Nicolas A. Menzies.

**Writing – review & editing:** Allison Portnoy, Rebecca A. Clark, Matthew Quaife, Chathika K. Weerasuriya, Christinah Mukandavire, Roel Bakker, Arminder K. Deol, Shelly Malhotra, Nebiat Gebreselassie, Matteo Zignol, So Yoon Sim, Raymond C. W. Hutubessy, Inés Garcia Baena, Nobuyuki Nishikiori, Mark Jit, Richard G. White, Nicolas A. Menzies.

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
