## [Editor Report · Decision Letter 0]

31 May 2022

Dear Dr Portnoy, 

Thank you for submitting your manuscript entitled "The cost and cost-effectiveness of novel tuberculosis vaccines in low- and middle-income countries: a modelling study" for consideration by PLOS Medicine.

Your manuscript has now been evaluated by the PLOS Medicine editorial staff and I am writing to let you know that we would like to send your submission out for external peer review.

Please re-submit your manuscript within two working days, i.e. by Jun 02 2022 11:59PM.

Kind regards,

Beryne Odeny

PLOS Medicine

---

## [Decision Letter · Decision Letter 1]

28 Jul 2022

Dear Dr. Portnoy,

Thank you very much for submitting your manuscript "The cost and cost-effectiveness of novel tuberculosis vaccines in low- and middle-income countries: a modelling study" (PMEDICINE-D-22-01725R1) for consideration at PLOS Medicine. 

[LINK]

In light of these reviews, I am afraid that we will not be able to accept the manuscript for publication in the journal in its current form, but we would like to consider a revised version that addresses the reviewers' and editors' comments. Obviously we cannot make any decision about publication until we have seen the revised manuscript and your response, and we plan to seek re-review by one or more of the reviewers. 

We expect to receive your revised manuscript by Aug 18 2022 11:59PM. Please email us (plosmedicine@plos.org) if you have any questions or concerns.

We look forward to receiving your revised manuscript. 

Sincerely,

Beryne Odeny, 

PLOS Medicine

plosmedicine.org

Comments from the Academic Editor:

I am in agreement with all of Reviewer #2’s comments and suggestions. They should provide more detail about what they did even if they do not reproduce the whole epi model. It is not reasonable to expect people to have to read their other papers to understand this one.

I did have a couple of other concerns. The hypothetical vaccine they model has attributes that are very similar to BCG which is now widely used in almost all the countries covered. Specifically, vaccine efficacy in newborns is around 72% in neonates and 47% in adolescents and adults and the durability is around 10 years although this latter is controversial. The Clark model acknowledges high BCG coverage among neonates in most high burden countries but does not specifically integrate that into the model. They note that they envision the new vaccine would be in addition to BCG at birth but don’t mention whether they consider BCG revaccination. Since the proposed efficacy of the new hypothetical vaccine in adolescents/adults is 50%, very similar to the efficacy of BCG in the Nemes South Africa clinical trial, the question arises of why not just use BCG at a tiny fraction of the cost. Presumably, the answer to that is that BCG revaccination efficacy has been highly heterogeneous in different areas – but that should serve as a warning that other TB vaccines may behave similarly depending on their biologic mechanism. It is also quite possible that children who "fail" BCG (ie. those who stand to benefit from another vaccine) are precisely those who might fail the new vaccine.

Another minor procedural point. The authors note that the funder (WHO) had no role in the study but at least one of the authors, Matteo Zignol, is from WHO.

1) Please confirm whether the underlying model has been submitted for publication elsewhere. Ideally, this model needs to be peer reviewed and published.

2) Abstract:

a) Under methods, and findings, please summarize the data sources for key estimates

b) Please ensure that all numbers presented in the abstract are present and identical to numbers presented in the main manuscript text.

c) In the last sentence of the Abstract Methods and Findings section, please describe the main limitation(s) of the study's methodology.

4) Did your study have a prospective protocol or analysis plan? Please state this (either way) early in the Methods section. 

5) Please ensure that the study is reported according to the CHEERS 2022 checklist and include the completed checklist as Supporting Information. Please add the following statement, or similar, to the Methods: "This study is reported as per the Consolidated Health Economic Evaluation Reporting Standards 2022 (CHEERS 2022) statement (S1 Checklist)." Please access this link for more details on the checklist: https://bmcmedicine.biomedcentral.com/articles/10.1186/s12916-021-02204-0

6) Please ensure you provide definitions for all abbreviations in your tables including USD

7) References:

a) Please select the PLOS Medicine reference style in your citation manager. In-text reference call outs should be presented as follows noting the absence of spaces within the square brackets, e.g., “…population [1,2].”

b) References should have no more and no less than six names before “et al.” 

c) Please ensure that weblinks are current and accessible to date.

8) Please remove the ‘Financial disclosure” and “competing interest,” statements from the main text. In the event of publication, this information will be published as metadata based on your responses to the submission form.

Comments from the reviewers:

Reviewer #1: This study estimated the cost and cost-effectiveness, and net monetary benefit of new TB vaccine introduction, from health system and societal perspectives using the WHO Full Value of Vaccines Assessment framework. This research topic is of importance and the results of the study can inform global-level decision-making for novel TB vaccine investment and introduction. However, I think the method section of the study is not clearly and sufficiently described. Below are my specific comments.

1. Please provide line numbers for easier reference. 

2. Method, Analytic overview: How did you select the starting year of 2028? Why did you select 2028 instead of 2023 or 2030 as the starting year?

3. Method, Vaccination scenarios, page 5: "we assumed an average ten-year duration of protection, with exponential waning." Do you have any reference/evidence for the assumption of ten-year protection and the exponential waning?

4. Method, Vaccination scenarios, page 5: What is definition of the routine vaccination program? Do you assume the routine vaccination the same across all 105 LMICs? I think there may be huge variations between countries regarding their routine vaccination programs. 

5. Method, Vaccination scenarios, page 5-6: "the adolescent/adult vaccine delivered through routine vaccination of nine-year-olds plus a one-time vaccination campaign for ages 10+." Why did you choose a routine vaccination of nine-year-olds? Is a nine-year-olds routine vaccination common in most LMICs? Also, the name "adolescent/adult vaccine" seems a bit misleading given that a big part of the program is vaccination among children of 9 years old.

6. Method, Vaccination scenarios, page 6: "stakeholders, we assumed a coverage target of 85% for the infant vaccine (average coverage of diphtheria-tetanus-pertussis third dose for LMICs11), 80% for routine delivery of adolescent/adult vaccine, and 70% of the adolescent/adult vaccination campaign." Assuming that all the 105 countries can achieve the same level of coverage with the same speed of reaching the target without considering the health system capacity of each country individually is highly unrealistic to me.

7. Method, Vaccination scenarios, page 6: "We assumed country-specific introduction years from 2028-2047, determined based on indicators for disease burden, immunization capacity, classification of the country as an "early adopter/leader," lack of regulatory barriers, and commercial prioritization." How did you determine the introduction years based on this factors? I think the authors need to elaborate a bit more. Also, if you already considered the differences between countries regarding these factors, how could you still assume all countries will reach the same level of coverage with the same speed?

8. Method, Epidemiological outcomes and health service utilization, page 6: "…data and 10 due to unsuccessful calibration results." What are the criteria for unsuccessful calibration results? The authors need to elaborate. 

9. Method, Epidemiological outcomes and health service utilization, page 6: "In countries with a significant burden of HIV-associated TB.." what is the definition of countries with a significant burden of HIV-associated TB. You need to be more specific about it.

10. Method, Cost outcomes, page 7: "Diagnostics costs for drug-susceptible (DS) and rifampicin-resistant (RR) TB were obtained from published literature, and extrapolated to all LMICs by country income level." Do you mean you adjusted the average diagnostics costs from literature by each country's income level? If so, how did you do the adjustment? Need to elaborate.

11. Method, Cost outcomes, page 7: "…we derived unit costs by extrapolating estimates reported by the Global Health Cost Consortium…" what did you mean by "extrapolating" and how did you do the extrapolation?

12. Method, Cost-effectiveness analysis, page 8: "We also reported a specification in which costs are discounted but not health outcomes." I personally think that health outcomes should not be discounted. I would recommend making this the main assumption.

13. Method, Statistical analysis, page 9: "We constructed probability distributions representing uncertainty in economic inputs and disability weights…" How did you construct the probability distributions? Did you assume normal distribution for those inputs? How did you determine the parameters of the distributions? I think the authors need to elaborate a bit more. 

Reviewer #2: Thank you for the opportunity to review this comprehensive and well written analysis. It includes important information about the potential cost impact and cost-effectiveness of novel TB vaccines. It focuses on an improved infant vaccine (80% efficacy), and an adolescent/adult vaccine with 50% efficacy in both previously infected and uninfected persons. The methods and results are wide-ranging, covering 105 countries, and buttressed by many sensitivity and scenario analyses.

Major comments:

1. The economic analysis builds on extensive epidemiologic modeling, the methods for and results of which are detailed in a separate manuscript available as a medRxiv preprint (and supporting supplements). While it would not be appropriate to repeat the same material in the economic analysis manuscript, some additional, reasonably high-level information about the epidemiologic modeling methods would be appropriate in the main text of the economic analysis manuscript, and especially in its supplement. For example, the figure showing the basic model structure for the dynamic epidemiologic model could be added to the supplement to the present manuscript. Ideally there would be sufficient information in the main text for readers to understand the broad outlines of the epidemiologic model, without having to refer elsewhere.

2. The major assumptions, parameters and values around key cost inputs for TB care need to be much better articulated. Again, I recognize that there may be extensive information in the references provided, but readers should not need to consult these for basic understanding. For example, it would seem appropriate to add to the supplement a table where key component cost inputs used for DS- and RR-TB care are provided for each of the 105 countries, including point estimates, ranges, and distributions (referred to in the first paragraph on p. 9, but not explained elsewhere). The same holds true for costs borne by patients and families. A high-level summary of these input health care and patient/family costs should also be provided in the main text, potentially grouped by country income group (e.g. low, low-middle, and high-middle income countries).

3. The authors should better defend/justify their primary focus on the 1x per-capita GDP per DALY averted threshold, as opposed to the alternatives they explore in sensitivity analysis. In addition, the "Best Buy $100" threshold is shown in the supplementary figures but not mentioned or defined in the main text. 

4. Along similar lines, I am somewhat uncomfortable with the restriction of the incremental net monetary benefit analysis to countries where the ICERs fall below the relevant cost threshold per DALY averted. I think at minimum the authors should add to the supplement a couple of tables showing estimated iNMB for the 105 countries as a whole. Otherwise the analysis might be taken to imply that adoption of the vaccines should be limited to those countries where they appear cost-effective based on those somewhat arbitrary thresholds. I suspect this is not the authors' intent, and indeed in the discussion it would be relevant to discuss their use of such thresholds with a policy lens. 

5. Perhaps my biggest concern about this manuscript relates to the novelty and impact of its key findings for policymakers and other stakeholders. Members of the author group are well aware of, reference (and in many cases contributed to) previous analyses that have addressed the potential epidemiologic and economic impact of novel TB vaccines. The authors need to more clearly articulate how their current findings fit in the context of previously published work in this area, including what is different/striking/more robust about their analysis and results—beyond the obvious fact that it covers a very large and diverse set of countries. The authors should also emphasize (at least in their discussion) the findings which they believe will further convince readers and stakeholders of the necessity and value of improved TB vaccines, beyond what is already known in this regard.

Minor comments:

1. It is not clear how this analysis, labeled as being conducted within the WHO Full Value of Vaccines Assessment Framework, differs methodologically from other analyses that simply considered both health system and societal perspectives.

2. Results, p. 10: please specify more clearly what are undiscounted vs. discounted costs.

3. In the Supplement, please provide the appropriate citation and reference for the disability weights shown in table S2.

Reviewer #3: Thank you for the opportunity to revise this great and important paper for the field. 

Find below some minor comments

Introduction. Initial sentence will be false in a matter of months. So, I would probably not rank the diseases in terms of mortality.

Methods. What was the scenario of a POI+POD vaccine in adults chosen and not consider them separately (given that it seems vaccines in pipeline are designed with the purpose of preventing either infection (which would indeed protect against disease) or just disease. In other words, in adults, you would likely need to administer POI+POD (two separate vaccines) and this would have cost implications. Of course, there might be POI candidates that work immunologically similarly as POD, but the effect against POD and POI might be different. In addition, I do not think this is how they are currently conceived.

It would be great to know what "lack of critical calibration data or unsuccessful calibration results" means?

If risk of TB goes up in most countries after 15 years of age, why was the age of 9 y-old selected? If we assume the efficacy wanes (as you do), you probably want to vaccine very close to the age where the risk greatly increases.

Is there any booster to the initial vaccination in children? The text needs clarification.

Should you specify in the text the HBC country in which infant vaccination would not be cost effective?

What about considering in the sensitivity analysis several vaccine doses? I think that several doses might be required (as it happens with many vaccines). Otherwise, I would suggest to state clearly the analysis is based on a single shot and perhaps acknowledge so in the limitations.

How do the costs and cost effectiveness compare to other vaccines (or similar models for other diseases, i.e malaria,hiv, pneumococcal disease ) Could you make a reasonable comparison?

[LINK]

---

## [Decision Letter · Decision Letter 2]

8 Nov 2022

Dear Dr. Portnoy,

Thank you very much for re-submitting your manuscript "The cost and cost-effectiveness of novel tuberculosis vaccines in low- and middle-income countries: a modelling study" (PMEDICINE-D-22-01725R2) for review by PLOS Medicine.

I have discussed the paper with my colleagues and the academic editor and it was also seen again by three reviewers. I am pleased to say that provided the remaining editorial and production issues are dealt with we are planning to accept the paper for publication in the journal.

[LINK]

We look forward to receiving the revised manuscript by Nov 15 2022 11:59PM.   

Sincerely,

Philippa Dodd, MBBS MRCP PhD

PLOS Medicine

plosmedicine.org

Requests from Editors:

Thank you for your considerate and detailed responses to previous editor and reviewer comments. Below are some further suggestions/revisions which we request you respond to in full.

COMPETING INTERESTS STATEMENT

You state in your financial disclosure that members of the funding body are also co-authors on the manuscript. Please update your COI statement to reflect this. Please include details of which co-authors (initials shall suffice) are members of the funding body and what each author’s role in the study was, within this statement.

AUTHOR SUMMARY

“tuberculosis disease” suggest avoiding use of the term “disease” perhaps “tuberculosis infection", or something similar - please check throughout and amend where necessary

FIGURES 

Please ensure that colour schemes used are accessible to those with colour blindness (i.e. avoiding the use of green or red)

DISCUSSION

Line 338: there is an additional line space separating the text, please correct

REFERENCES

Where more than one reference is cited, please remove the spaces separating the citations. For example, line 17: [5, 6] should read [5,6]

Comments from Reviewers:

Reviewer #1: I am mostly satisfied with the responses and revisions made by the authors. 

The only concern I still have is the assumption that all 105 countries can achieve the same level of vaccine coverage with the same speed of reaching the target. There could be some simple way to account for the capacity of health system for each country. For example, the HAQ index produced by IHME might be used to account for different health system capacity in each country, which could potentially make the results more realistic and useful to policy makers from different countries. 

Reviewer #2: Thank you for your careful and extensive revisions to your manuscript, which I think have more than adequately addressed my and other reviewers' comments.

I have two minor comments/suggestions:

1. In your abstract, please clearly indicate that the modeled vaccines were for prevention of disease (i.e. not prevention of infection).

2. It is a foregone conclusion that any analysis that discounts costs, but not health outcomes, will make any intervention appear "more" cost-effective than the corresponding analysis which discounts both. Hence I think you can remove general statements to this effect (e.g. p, 17, line 297). I know you are well aware of (and cite) the methodologic reasons for discounting both health outcomes and costs for purposes of CEA. There may be good reasons to report undiscounted health outcomes alone, and sometimes a fully undiscounted CEA as a sensitivity analysis (which can then be compared with the base case results). However, undue emphasis on an analysis which discounts only costs can be problematic. Certainly I think one should avoid making any substantive conclusions on this basis.

Thank you again for the extensive work done.

Reviewer #3: Thank you for addressing my comments, well done.

[LINK]

---

## [Decision Letter · Decision Letter 3]

9 Dec 2022

Dear Dr Portnoy, 

On behalf of my colleagues and the Academic Editor, Dr. Megan Murray, I am pleased to inform you that we have agreed to publish your manuscript "The cost and cost-effectiveness of novel tuberculosis vaccines in low- and middle-income countries: a modelling study" (PMEDICINE-D-22-01725R3) in PLOS Medicine.

PRESS

Best wishes,

Pippa 

Philippa Dodd, MBBS MRCP PhD 

PLOS Medicine